# Sequencing of the Viral UL111a Gene Directly from Clinical Specimens Reveals Variants of HCMV-Encoded IL-10 That Are Associated with Altered Immune Responses to HCMV

**DOI:** 10.3390/ijms23094644

**Published:** 2022-04-22

**Authors:** Shelley Waters, Silvia Lee, Ibnu Ariyanto, Nina Kresoje, Shay Leary, Kylie Munyard, Silvana Gaudieri, Ashley Irish, Anthony D. Keil, Richard J. N. Allcock, Patricia Price

**Affiliations:** 1Curtin Medical School, Curtin Health Innovation Research Institute, Curtin University, Bentley 6102, Australia; shelley.waters@postgrad.curtin.edu.au (S.W.); silvia.lee@curtin.edu.au (S.L.); k.munyard@exchange.curtin.edu.au (K.M.); 2PathWest Laboratory Medicine WA, Department of Microbiology, Nedlands 6009, Australia; anthony.keil@health.wa.gov.au; 3Virology and Cancer Pathobiology Research Center, Faculty of Medicine, Universitas Indonesia, Jakarta 10430, Indonesia; ibnu.ariyanto07@gmail.com; 4School of Biomedical Sciences, University of Western Australia, Nedlands 6009, Australia; nina.kresoje@uwa.edu.au (N.K.); richard.allcock@uwa.edu.au (R.J.N.A.); 5Institute for Immunology and Infectious Diseases, Murdoch University, Murdoch 6150, Australia; s.leary@iiid.murdoch.edu.au (S.L.); silvana.gaudieri@uwa.edu.au (S.G.); 6School of Human Sciences, University of Western Australia, Nedlands 6009, Australia; 7Department of Medicine, Division of Infectious Diseases, Vanderbilt University Medical Center, Nashville, TN 37232, USA; 8Department of Nephrology, Fiona Stanley Hospital, Murdoch 6150, Australia; ashley.irish@health.wa.gov.au; 9PathWest Laboratory Medicine WA, Department of Diagnostic Genomics, Nedlands 6009, Australia

**Keywords:** human cytomegalovirus, interleukin-10, UL111a, renal transplant recipients, people with HIV, deep sequencing

## Abstract

Human cytomegalovirus (HCMV) is a beta-herpesvirus carried by ~80% of adults worldwide. Acute infections are often asymptomatic in healthy individuals but generate diverse syndromes in neonates, renal transplant recipients (RTR), and people with HIV (PWH). The HCMV gene UL111a encodes a homolog of human interleukin-10 (IL-10) that interacts with the human IL-10 receptor. Deep sequencing technologies were used to sequence UL111a directly from 59 clinical samples from Indonesian PWH and Australian RTR, healthy adults, and neonates. Overall, 93% of samples contained more than one variant of HCMV, as defined by at least one nonsynonymous variation. Carriage of these variants differed between neonates and adults, Australians and Indonesians, and between saliva and blood leukocytes. The variant alleles of N41D and S71Y occurred together in Australian RTR and were associated with higher T-cell responses to HCMV pp65. The variant P122S was associated with lower levels of antibodies reactive with a lysate of HCMV-infected fibroblasts. L174F was associated with increased levels of antibodies reactive with HCMV lysate, immediate-early 1 (IE-1), and glycoprotein B (gB) in Australian RTR and Indonesians PWH, suggesting a higher viral burden. We conclude that variants of UL111a are common in all populations and may influence systemic responses to HCMV.

## 1. Introduction

More than 80% of adults worldwide are seropositive for Human Cytomegalovirus (HCMV) [1]. Acute infections are generally asymptomatic in healthy individuals. However, HCMV seropositivity is associated with accelerated development of cardiovascular disease (CVD) [2], so early markers of CVD such as flow-mediated dilatation (FMD) may be considered as a clinical footprint of HCMV infection [3]. 

In neonates, HCMV infection commonly causes sensorineural hearing loss and carries increased risks of morbidity, neurodevelopment delays, and vision impairment due to central nervous system damage [4]. HCMV infections occur in 20–60% of organ transplant recipients and are associated with rejection, secondary bacterial or fungal infections, and CVD. The level of risk that HCMV poses can be influenced by the type of organ transplanted, immunosuppressive medications, and prophylactic regimen [5]. Recipients who are HCMV-seronegative risk severe complications from primary HCMV infection, including graft rejection events and mortality [6].

Most people with HIV (PWH) are HCMV-seropositive [7,8,9]. HCMV retinitis is considered an AIDS-defining illness and can lead to blindness [10]. Although retinitis is now rare, PWH on antiretroviral therapy (ART) maintain higher levels of circulating HCMV-reactive antibodies than the general population [11]. Higher antibody levels are associated with accelerated CVD and cerebrovascular disease [12].

HCMV has a large genome of around 235 kb [13], encoding 165–252 open reading frames (ORFs). However, only 45 ORFs are essential for replication in vitro, with the remainder of the genome involved in immunomodulation [14,15,16,17]. Several immunomodulatory genes are homologs of host genes. This includes UL111a, which encodes homologs of human interleukin-10 (IL-10). 

HCMV UL111a is differentially spliced creating several isoforms [18]. The transcripts cmvIL-10 and LAcmvIL-10 are the best characterized. They differ in length and number of exons [19]. It is accepted that HCMV establishes latency in fixed tissues where active replication and viral progeny cannot be detected. However, in vitro studies have shown that numerous viral genes are continually expressed during the latent state. UL111a continues to be expressed during latency but only the LAcmvIL-10 transcript is detectable [20].

Both cmvIL-10 and LAcmvIL-10 are broadly immunosuppressive [19]. cmvIL-10 can interact with human IL-10R1 and initiate signaling via STAT-3 [21]. This can modulate cellular IL-10 synthesis. LAcmvIL-10 has more restricted functions which include downregulation of surface major histocompatibility complex (MHC)-II, and reducing antigen presentation during latency [22].

Several studies have examined HCMV diversity via Sanger sequencing of PCR amplicons. However, this approach probably misses mixed-strain infections, which are relatively common [23,24,25]. Other studies have sequenced HCMV propagated *in vitro*, potentially missing variants present only *in vivo*. Here, we describe a nested PCR protocol with deep sequencing technology applied directly to clinical samples. We present UL111a gene sequences from RTR, PWH, healthy adults, and neonates. UL111a sequences are compared with the Toledo reference strain that was derived from the urine of a congenitally HCMV-infected child [26]. We utilized an Australian RTR cohort recruited more than 2 years after transplantation [27] and the JakCCANDO project, which provides a longitudinal cohort of PWH commencing ART, and followed over 12 months in Jakarta, Indonesia [28].

## 2. Results

### 2.1. Most Clinical Samples Contain More Than One Variant of HCMV

Amplicon sequences targeting HCMV UL111a were obtained from 59 clinical samples (blood, saliva, or urine), with a mean depth of 11,734. In total, 27 samples were from Indonesian PWH (21 buffy coat and 6 saliva) collected after 0 to 3 months on ART, 21 were from Australian RTR (>2 years after transplant; 8 buffy coat and 13 saliva), 7 were from Australian healthy adults (2 buffy coat and 5 saliva), and 4 were from Australian neonates (urine).

Compared with the Toledo reference strain (GenBank no. GU937742.1), there were 311 sites of nucleotide variation (Figure 1a). Hereafter, we use the term “variant” to define nonsynonymous changes, as they are the focus of this study. There were 32 sites of variation that were present in 3 or more individuals (Figure 1b). Of the 59 samples sequenced, 55 samples (93%) contained more than one variant of HCMV, based on the presence of variations in UL111a. Four of the five samples with a single strain differed from the Toledo protein sequence. The carriage of the variant sequences is discussed below. Position numbers relate to the cmvIL-10 transcript (*aka* Transcript “A”).

### 2.2. Several Polymorphisms Were Group-Specific

UL111a sequences from neonates (n = 4) had five variants, none of which were unique to this group. While sequences from adults (n = 55) had 32 variants (Table 1), of which 27 were only observed in adults. UL111a sequences from Australian adults had 29 variants, including 3 unique to Australian samples. HCMV sequences from Indonesians had 29 variants, including 3 unique to Indonesian samples. UL111a sequences from buffy coat samples had 32 variants, including 3 not found in saliva. Sequences from saliva had 29 variants, with none unique to saliva. Variants S73Y, K114E, P122S, Q132* (* denotes a stop codon) and L174F were present in all groups and all sample types. K114E was the most frequent and was present in 93% (55/59) of samples and was present in single strain samples. The null allele Q132* was present in 78% (46/59) of samples. Four samples carried only the Q132* stop codon, but this included three individuals in whom a second sample carried the Q allele (one RTR and 2 HIV patients). The fourth example was a congenital case.

### 2.3. Amino Acid Haplotypes Differ between Samples from Australia and Indonesia

Of the 32 variants, 16 were biallelic and were included in haplotyping analyses (Table 2). We included the triallelic locus L174F/W because it was associated with systemic responses to HCMV. The W allele occurred in 4 samples, which were grouped with the 14 carriers of the F allele for the haplotype analysis. We identified 12 haplotypes (numbered UL111a-1 to UL111a-12), which occurred in 2 or more samples and accounted for 78% of all genotypes. UL111a-2 was more frequent in Indonesian than Australian samples (14/27 versus 5/28; *p* = 0.01). The minor alleles at positions 9, 123, 127, 184, 189, and 214 did not occur in any haplotypes found in ≥2 samples. The variant K114E was found in all 12 haplotypes. Haplotype UL111a-7 was restricted to Australian adults (*p* = 0.05).

### 2.4. UL111a Variations Are Associated with Levels of HCMV-Reactive Antibody

HCMV encoding N at position 41 (D41N) was only present in samples from Australian adults (5/28 samples), including 3/15 RTRs and 1/6 healthy adults (who had HCMV sequenced in both saliva and buffy coat). All cases with those who carried N at position 41 also carried the minor allele of S73Y in haplotype UL111a-7. RTR carrying the N allele had higher T-cell responses to HCMV pp65 (Figure 2A, *p* = 0.03) and slightly higher proportions of Vδ2^−^ γδ T cells (Figure 2B, *p* = 0.08). This population has been linked with HCMV seropositivity and disease [29]. Other variants within the haplotype may contribute to the phenotype.

HCMV encoding S at P122S was present in all cohorts (14/59 samples), including 4/15 RTR. RTR carrying P122S had lower levels of antibodies reactive with a lysate of HCMV-infected fibroblasts (Figure 3A, *p* = 0.03), and slightly lower levels of HCMV IE-1-reactive antibody (Figure 3B, *p* = 0.08). The same individuals had inferior vascular endothelial function when assessed by FMD (Figure 3C, *p* = 0.049).

HCMV encoding F at position 174 was present in Australian and Indonesian adults but not in neonates (14/55 samples). This included 6/15 Australian RTR and 7/18 Indonesian PWH. This analysis excludes two PWH carrying only the 174W variant but includes one RTR encoding both L and W. This genotype was also found in one healthy control. Amongst RTR, 174F was associated with elevated levels of HCMV gB-reactive antibody (Figure 4A, *p* = 0.02). In Indonesian PWH, 174F was associated with elevated levels of antibody reactive with HCMV lysate (Figure 4B, *p* = 0.046) or HCMV IE-1 (Figure 4C, *p* = 0.03) before ART. The trends remained after 1 month on ART.

## 3. Discussion

Few studies have sought HCMV sequences obtained from deep sequencing directly from clinical samples. Suárez et al. (2019) utilized high-throughput sequencing of enriched DNA libraries produced directly from clinical samples, which provided insight into HCMV gene recombination [30]. We have previously published deep sequencing of the HCMV gene US28 directly from clinical samples [25] and demonstrated variants that were associated with host responses. U111a variants from clinical samples have not previously been linked with clinical manifestations or outcomes. Here, we used a nested PCR, followed by a deep sequencing approach, on HCMV directly from clinical samples from PWH, RTR, healthy controls, and neonates, to study variations in UL111a. Three seronegative individuals (1 RTR and 2 controls) yielded UL111a sequences, but these were not distinguishable as a group [31]. Multi-variant infections were common (55/59 samples). The four available neonatal samples all displayed mixed infections but revealed fewer sites of variation than were seen in adults and none were unique to neonates (Table 1). Two cases were asymptomatic—one had hepatitis that resolved spontaneously, and one had sensorineural hearing loss. This infant had a single variant carrying only 174F. Hence, we cannot associate any individual variation with clinical outcomes in neonates. In contrast, when US28 was analyzed in neonates, only two of the four samples had multiple US28 variants [25]. While some studies suggest multi-variant infections are relatively rare [32], our data supports the notion that they are common [33]. Clinically, multi-variant infections are associated with ganciclovir resistance and graft rejection in organ transplant recipients [34,35].

The stop codon at Q132* is likely to modify the structure of the encoded protein as the downstream sequence encodes several cysteine residues. However, the truncated protein may retain properties resembling LAcmvIL-10, which terminates around amino acid 139 [36]. It may be significant that this only existed as a unique sequence in a congenital sample.

Haplotype analyses can provide an estimate of the age of variation where fixed combinations suggest more ancient mutation events. Therefore, predicting which variants are carried and maintained through transmission. The variant K114E was included in all haplotypes, suggesting that this variant is prevalent in clinical samples, when compared with laboratory strains such as Toledo. Here, haplotype UL111a-2 differed from Toledo at P83L, and K114E and was more common in samples from Indonesia than in those from Australia. Haplotype UL111a-7 differed from Toledo at D41N, S73Y, and K114E and was more common in samples from Australia than in those from Indonesia. In Australian samples, D41N and S73Y were always carried together. The results suggest that different viruses circulate in Indonesia and Australia.

The long-term goal of studies such as ours is the prediction of disease manifestation and clinical outcomes as a result of HMCV infection. Here, we present the first steps towards this goal using minor alleles carried in UL111a. Future studies will correct for the presence of other mutations in host and viral genes, co-morbidities, and socio-demographic factors. This could begin with D41N and/or S73Y, P122S, and L174F, which were associated with HCMV-reactive antibody levels in the cohorts described here. This may reflect modulation of the burden of HCMV or of the induction of the immune responses measured.

The cmvIL-10 protein reduces the expression of MHC-I and MHC-II by cultured monocytes and the generation of interferon-γ and other pro-inflammatory cytokines [37]. D41N was always carried with S73Y in Australian RTR (haplotype UL111a-7) and associated with elevated HCMV pp65 specific T cells producing interferon-γ. Accordingly, frequencies of HCMV-induced Vδ2^−^ γδ T cells were marginally higher in individuals carrying D41N. It is plausible that cmvIL-10 encoded with D41N or S73Y may be less efficient at suppressing T-cell activation.

The L174F variant was more common than the other variants assessed (14/55 samples/people). L174F is associated with elevated levels of gB-reactive antibodies in RTR, and with elevated levels of HCMV lysate and IE-1 reactive antibodies in PWH before ART. There is evidence that HCMV IL-10 can stimulate B-cell proliferation [38], so the F variant may be more active in this regard.

RTR carrying the P122S variant had lower levels of HCMV lysate-reactive antibodies and marginally lower IE-1-reactive antibodies. This suggests that P122S may be involved in more effective suppression of antigen presentation or less effective stimulation of B-cell proliferation. Furthermore, 122S was not carried with 41N or 174F in haplotype analyses (Table 2), and it was also associated with lower FMD, marking inferior vascular function. HCMV-reactive antibodies correlated inversely with FMD in both PWH and RTR, suggesting a relationship between FMD and the burden of HCMV [39,40]. The association between 112S and FMD supports this finding.

## 4. Materials and Methods

### 4.1. RTR and Healthy Controls from Perth, Western Australia

In total, 82 RTR were recruited from renal clinics at Royal Perth Hospital (Western Australia). Inclusion criteria were clinical stability greater than two years after transplant (median (range) 7 (2–37) years), no HCMV disease or reactivation within six months of sample collection, and no current antiviral treatment. RTR infected with hepatitis B or C were excluded. At recruitment, 71/84 patients were HCMV-seropositive, with a median age of 57 (31–76) years). Our assessments of seropositivity concurred with those made at the time of transplantation, so we were not assessing primary infections. Healthy adult controls were recruited in parallel. Among them, 49/81 were HCMV-seropositive with a median age of 55 (21–86) years. Ethics approval was obtained from Royal Perth Hospital Human Research Ethics Committee (approval number: EC 2012/155) and endorsed by Curtin University Human Research Ethics Committee (approval numbers: HRE16-2015 and HRE2021-0044). Participants provided written informed consent [27].

### 4.2. People with HIV from Jakarta, Indonesia

The JakCCANDO Project is a comprehensive survey of clinical and immunological responses to ART undertaken in Cipto Mangunkusumo Hospital’s outpatient clinic (Jakarta, Indonesia) [28]. In total, 82 ART-naïve PWH were enrolled during 2013–2014, with <200 CD4 T cells/µL. The study was approved by Universitas Indonesia, Cipto Mangunkusumo Hospital, and Curtin University ethics committees. Written informed consent was obtained from each subject. Samples were collected before ART initiation (V0) and at months 1, 3, 6, and 12 (V1, V3, V6, V12).

### 4.3. Australia Neonates

Four de-identified congenital urine samples in virus transport media were provided by the Department of Microbiology, PathWest Laboratory Medicine WA. Samples were collected between 1 and 13 days of life and all four had detectable HCMV DNA when assessed by routine hospital assays. Two neonates had symptomatic infections. One had hepatitis attributed to HCMV that spontaneously resolved without antiviral therapy. Another had bilateral sensorineural hearing loss, other central nervous system and lymphatic abnormalities, and required antiviral therapy.

### 4.4. Extraction and Detection of HCMV DNA

DNA was extracted from saliva, buffy coat, or urine using FavorPrep Blood Genomic DNA Extraction Mini Kits (Favorgen, Ping-Tung, Taiwan) and stored at 80 °C. HCMV was detected using an in-house qPCR assay with primers targeting the UL54 gene (HCMV DNA polymerase) [41]. Samples positive via UL54 qPCR were selected for sequencing.

### 4.5. Targeted Whole Gene Amplification

Primers targeting UL111a were designed using Geneious 8.1.9 (https://www.geneious.com) (5′-3′: F-TTCGTCTT-GATCTCCAGCCG, R- GCAACACCCACAAACAACGT). Reactions were performed in a total volume of 20 µL containing 0.4 µL of MyTaq HS DNA polymerase (Bioline, Meridian Bioscience, Cincinnati, OH, USA), 4 µL of MyTaq reaction buffer, 0.8 µL of 10uM primers (Sigma-Aldrich, St. Louis, MI, USA), and 5 µL of DNA diluted 1:2. Cycling conditions were 1 min at 95 °C, followed by 30 cycles of 15 sec at 95 °C, 15 sec at 60 °C, and 1.5 min at 72 °C, followed by a final extension step of 7 min at 72 °C. Amplicons were purified prior to the preparation of DNA libraries using MO BIO Laboratories UltraClean PCR Clean-Up Kits (QIAGEN, Hilden, Germany).

### 4.6. Preparation of Ion Ampliseq™ DNA Libraries

Libraries were prepared using an Ion Ampliseq™ Library Kit 2.0 with halved reaction volumes and a total of 10 ng of template nucleic acid. The targets were amplified for 30 cycles with an anneal/extension time of 4 min per cycle. During library purification, ethanol was freshly prepared at 75% concentration. Libraries were quantified using a High-Sensitivity DNA Kit on a Bioanalyzer 2100 (Agilent, Santa Clara, CA, USA).

### 4.7. Libraries Were Sequenced Using an Ion Proton Sequencer

Barcoded sample libraries were diluted in low Tris-EDTA (Thermo Fisher Scientific, Waltham, MA, USA) to reach a final concentration of 100 pmol/L, and equal volumes of each were pooled. The pooled libraries then underwent template preparation on an Ion Chef System and were loaded onto Ion P1 v3 sequencing chips using an Ion PI Hi-Q Chef Kit (Thermo Fisher Scientific). Semiconductor sequencing was performed on an Ion Proton Sequencer (Thermo Fisher Scientific) using an Ion PI Hi-Q Sequencing Kit (Thermo Fisher Scientific) [42].

### 4.8. Immunological Assessments of HCMV

Plasmas stored at −80 °C were assessed for HCMV-reactive IgG titers using in-house ELISAs based on a lysate of fibroblasts infected with HCMV AD169, recombinant HCMV gB (Chiron Diagnostics, Medfield, MA, USA) or IE-1 protein (Miltenyi Biotech, Cologne, Germany). Results are presented as arbitrary units (AU)/mL based on a standard plasma pool, allowing comparisons between people but not between antigens [27,43].

Peripheral blood mononuclear cells (PBMCs) were used to assess T-cell responses (interferon-gamma production) to pp65 (JPT Peptide Technologies; Berlin, Germany) using an ELISPOT assay. These antigens are known to raise CD4 and CD8 T-cell responses. PBMC was also used to enumerate populations of Vδ2- γδ T cells using multicolor flow cytometry, as these are elevated in CMV-seropositive RTR [29].

### 4.9. Assessment of Vascular Pathology

Ultrasonography was used to assess FMD of the brachial artery after 10 min of rest in Australian RTR and healthy controls [44]. FMD assesses the ability of the larger conduit artery to respond to shear stress via endothelial-dependent and -independent mechanisms.

### 4.10. Data Analysis

Sequences were mapped to the Toledo reference (GenBank: GU937742.1) using the tmap tool within the Torrent Suite v 5.10. BAM files mapped to Toledo were loaded into proprietary software, Visual Genomics Analysis Suite (VGAS) (http://www.iiid.com.au/software/vgas) [42]. Variants were called if they occurred at a frequency of greater than 10% and had a minimum of 50 reads. VGAS was also utilized to predict changes in the protein sequence.

Amino acid haplotypes and their estimated frequencies were determined using the default parameters of the fastPHASE algorithm, with the exception that haplotypes were sampled an additional 5000 times [45]. Haplotypes with a population frequency of less than 1% were excluded from analyses. Haplotypes are labeled UL111a-1 to UL111a-12 in descending order of their frequencies.

### 4.11. Statistical Analyses

Continuous data were analyzed with Mann–Whitney non-parametric statistics, and categorical data were analyzed with Chi-squared or Fisher’s exact tests, as appropriate, using GraphPad Prism version 8 for Windows (GraphPad Software, La Jolla, CA, USA).

## 5. Conclusions

We demonstrated the diversity of UL111a in Australian RTR, healthy adults and neonates, and Indonesia PWH. Variants were shown to exist in haplotypes, which suggests that the variants are ancient and maintained when the virus is transmitted. We also showed preliminary evidence that the presence of some variants may influence the immunomodulatory functions of cmvIL-10. This is demonstrated by altered levels of HCMV-specific T cells and HCMV-reactive antibodies in the presence of those variants. Future studies should explore these variants while controlling for co-morbidities and other host or viral variations.

## Figures and Tables

**Figure 1 ijms-23-04644-f001:**
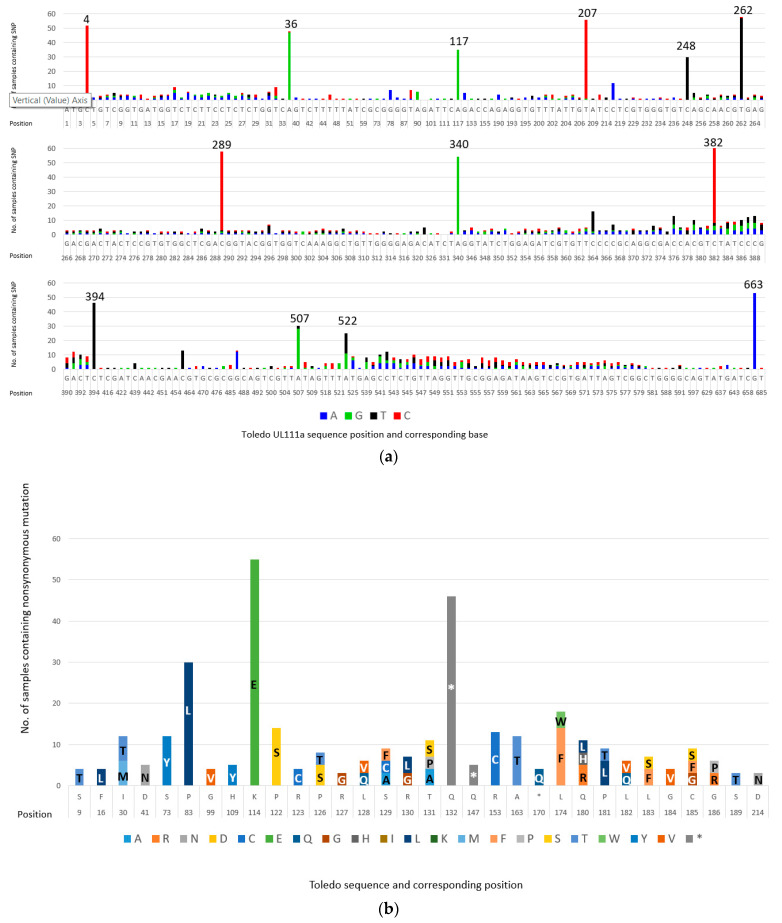
(**a**) **Summary of all nucleotide variations identified in HCMV sequenced in 59 samples.** Variations are displayed in reference to HCMV Toledo strain. Blue bars represent A, green bars represent G, black bars represent T, and red bars represent C. The height of the bars represents the number of samples the variation was found; (**b**) **Summary of all nonsynonymous mutations identified in HCMV sequenced in 59 samples.** Variations are displayed in reference to HCMV Toledo strain. Amino acids are represented by their one-letter codes. Each variation presented was found in at least 3 samples. The height of the bars represents the number of samples in which the variation was present.

**Figure 2 ijms-23-04644-f002:**
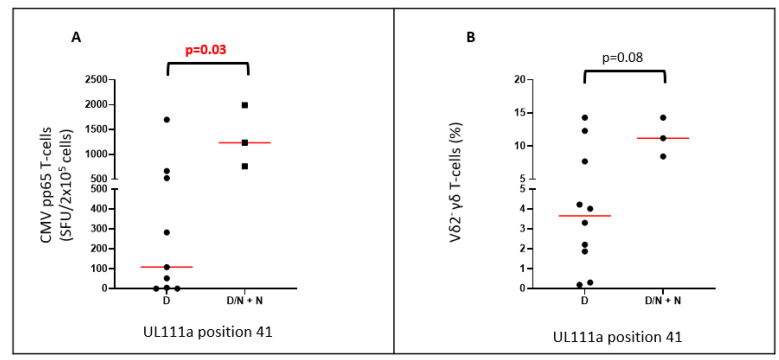
**RTR carrying HCMV with the D41N variant have more HCMV-reactive T cells and higher proportions of Vδ2^−^ γδ T-cells:** (**A**) comparison of HCMV pp65 specific T-cell responses in RTR carrying HCMV with D at position 41 and those with either D/N or N; (**B**) comparison of proportions of Vδ2^−^ γδ T cells in RTR carrying HCMV with only D at position 41 and those carrying D/N and only N.

**Figure 3 ijms-23-04644-f003:**
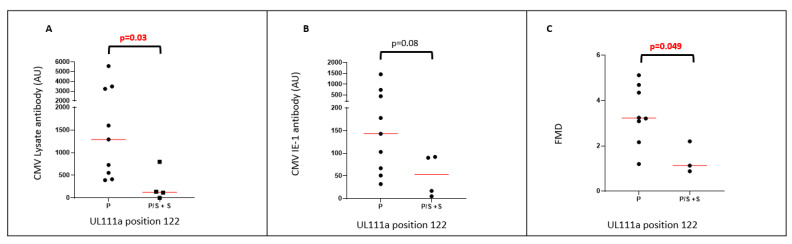
**RTR carrying HCMV with the P122S variant have lower levels of HCMV-reactive antibodies and decreased FMD:** (**A**) comparison of HCMV lysate-reactive antibodies in RTR carrying HCMV with P at position 122 and those carrying either P/S and S; (**B**) comparison of HCMV IE-1-reactive antibodies in RTR carrying HCMV with P at position 122 and those carrying either P/S and S; (**C**) comparison of FMD in RTR carrying HCMV with P at position 122 and those carrying either P/S and S.

**Figure 4 ijms-23-04644-f004:**
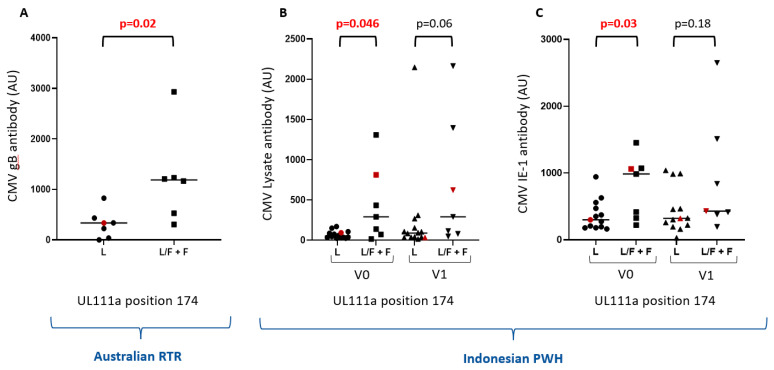
**RTR and PWH carrying HCMV with the L174F variant have higher levels of HCMV-reactive antibodies:** (**A**) comparison of HCMV gB-reactive antibodies in Australian RTR carrying HCMV with L at position 174 and those carrying either L/F or F; (**B**) comparison of HCMV lysate-reactive antibodies in Indonesian PWH carrying HCMV with L at position 174 and those carrying either L/F or F; (**C**) comparison of HCMV IE-1-reactive antibodies in Indonesian PWH carrying HCMV with L at position 174 and those carrying either L/F or F. V0 = baseline (0 months on ART), V1 = 1 month on ART. The red dots represent individuals also carrying L174W variant.

**Table 1 ijms-23-04644-t001:** UL111A protein variants distinct from Toledo were found in all groups.

Residue Position	Toledo Reference	Neonates n = 4	Adults n = 55	Australian n = 28	Indonesian n = 27	Buffy Coat n = 31	Saliva n = 24
9	S	S	S/**T**	S/T	S/T	S/T	S/T
16	F	F	F/**L**	F/**L**	F	F/L	F/L
30	I	I/T/M	I/T/M	I/**T**/**M**	I	I/T/M	I/T/M
41	D	D	D/**N**	D/**N**	D	D/N	D/N
73	S	S/Y	S/Y	S/Y	S/Y	S/Y	S/Y
83	P	P	P/**L**	P/L	P/L	P/L	P/L
99	G	G	G/**V**	G/V	G/V	G/V	G/V
109	H	H	H/**Y**	H/Y	H/Y	H/Y	H/Y
114	K	E	K/**E**	E/K	E/K	K/E	K/E
122	P	P/S	P/S	P/S	P/S	P/S	P/S
123	R	R	R/**C**	R/C	R/C	R/C	R/C
126	P	P	P/**T**/**S**	P/T/**S**	P/T	P/T/S	P/S/T
127	R	R	R/**G**	R/G	R/G	R/G	R/G
128	L	L	L/**Q**/**V**	L/Q/V	L/Q/V	L/Q/V	L/Q/V
129	S	S	S/**F**/**A**/**C**	S/F/A/C	S/F/A/C	S/F/A/C	S/F/A/C
130	R	R	R/**L**/**G**	R/L/G	R/L/G	R/L/G	R/L/G
131	T	T	T/**A**/**S**/**P**	T/A/S/P	T/A/S/P	T/A/S/P	T/A/S/P
132	Q	Q/*	Q/*	Q/*	Q/*	Q/*	Q/*
147	Q	Q	Q/*	Q/*	Q/*	Q/*	Q/*
153	R	R	R/C	R/C	R/C	R/C	R/C
163	A	A	A/T	A/T	A/T	A/T	A/T
170	*	*	*/**Q**	*/Q	*/Q	*/Q	*/Q
174	L	L/F	L/F	L/F/W	L/F/W	L/F/W	L/F/W
180	Q	Q	Q/**H**/**L**/**R**	Q/H/R	Q/H/**L**/R	Q/H/**L**/R	Q/H/R
181	P	P	P/**L**/**T**	P/L/T	P/L/T	P/L/T	P/L/T
182	L	L	L/**V**/**Q**	L/V/Q	L/V/Q	L/V/Q	L/V/Q
183	L	L	L/**F**/**S**	L/F	L/F/S	L/F/S	L/F
184	G	G	G/**V**	G/V	G/V	G/V	G/V
185	C	C	C/**G**/**S**/**F**	C/S/F	C/**G**/S/F	C/**G**/S/F	C/S/F
186	G	G	G/**P**/**R**	G	G/**P**/**R**	G/**P**/**R**	G
189	S	S	S/**T**	S	S/**T**	S/**T**	S
214	D	D	D/**N**	D	D/**N**	D/**N**	D

Nonsynonymous mutations are displayed in reference to Toledo. Changes unique to a group are in bold. All mutations reported were present in at least 3 samples. * denotes a stop codon.

**Table 2 ijms-23-04644-t002:** Haplotype UL111a-2 is frequently found in HCMV from Indonesian samples.

Position Toledo Variant	9	16	41	73	83	99	109	114	122	123	127	153	170	174	184	189	214	Indo (n = 27)	Aus (n = 28)	*p* ^a^
	S	F	D	S	P	G	H	K	P	R	R	R	*	L	G	S	D			
	T	L	N	Y	L	V	Y	E	S	C	G	C	Q	F/W	V	T	N			
**Haplotypes**																				
UL111a-1	S	F	D	S	P	G	H	E	P	R	R	R	*	L	G	S	D	12	12	0.99
UL111a-2	S	F	D	S	L	G	H	E	P	R	R	R	*	L	G	S	D	14	5	**0.01**
UL111a-3	S	F	D	S	P	G	H	E	P	R	R	R	*	F/W	G	S	D	3	3	0.99
UL111a-4	S	F	D	Y	P	G	H	E	P	R	R	R	*	L	G	S	D	2	3	0.99
UL111a-5	S	F	D	S	P	G	H	E	S	R	R	R	*	L	G	S	D	1	0	0.49
UL111a-6	S	F	D	S	L	V	H	E	P	R	R	R	*	L	G	S	D	1	1	0.99
UL111a-7	S	F	N	Y	P	G	H	E	P	R	R	R	*	L	G	S	D	0	5	**0.05**
UL111a-8	S	F	D	S	P	G	H	E	S	R	R	C	*	L	G	S	D	0	3	0.24
UL111a-9	S	F	D	S	L	G	H	E	P	R	R	R	Q	L	G	S	D	2	0	0.24
UL111a-10	S	L	D	S	P	G	H	E	S	R	R	C	*	L	G	S	D	0	2	0.49
UL111a-11	S	F	D	S	P	G	Y	E	S	R	R	C	*	L	G	S	D	1	1	0.99
UL111a-12	S	F	D	S	L	G	Y	E	P	R	R	R	*	L	G	S	D	1	1	0.99

^a^ Fisher’s Exact test comparing Australian and Indonesian adult samples (saliva or blood leukocytes), bold indicates that statistical significance was reached. Indo = Indonesian samples, Aus = Australian samples. Grey shading represents variations in comparison with Toledo reference. * denotes a stop codon.

## Data Availability

Amplicon sequence data have been deposited in NCBI under accession no. SAMN21506830 to SAMN21506889.

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
