# Peer review of "Sequencing of the Viral UL111a Gene Directly from Clinical Specimens Reveals Variants of HCMV-Encoded IL-10 That Are Associated with Altered Immune Responses to HCMV"

_ijms, 2022, doi:10.3390/ijms23094644_

Round 1
Reviewer 1 Report
The study by Waters et al., is well written and deals with an interesting question. However in my opinion the data do not allow to make any associations with regard of UL111a variants and an altered immune response.
Moreover there are some inconclusive and missing descriptions of the cohort and the data which makes it hard for the reader to evlauate the importance of their findings.
Here are some examples of my concerns:
- The same study cohort has already been used to investigate the US28 variants. (I) is not well understandable why the investigations of the 2 different genes have not been combined for interpretation. (ii) no clinical and demographic characteristics of the study cohort is given; e.g. it would be of interest to know the serostatus combinations of the RTRs, to know if the healthy adults are primary infected or not; the age of the patients could also have a severe influence on the Ab titres; the viral loads, when the samples were taken, etc.
- deep sequencing allows to reveal mixtures, however the mixed variant data have not been shown;
- it is not well explained why only 3 variant postions have been picked out;
- there is no information given for Figure 2 and Figure 3, which patients are exactly included into the comparing cohorts, no information is given on additional variables such as age, D/R status, viral load, etc.; moreover, it would be very helpful to see the T cell and antibody comparison between the total number of patients in each cohort;
Reviewer 2 Report
In the manuscript by Waters and co-authors HCMV encoded vIL-10 (UL111a) sequences from various populations are described. The vIL-10 has been associated with immunosuppression and reduced antigen presentation. Water et al. sequenced UL111a from cohort of people with HIV (Indonesia) and from a cohort of renal transplant patients and healthy controls (both from Australia). In addition, four samples from neonates (Australia) were included. In total, 59 samples were sequenced. Of these, 55 samples contained more than one nonsynonymous nucleotide change, resulting in a change in the protein sequence, here called variant. Three vIL-10 variants could be associated to immune response changes. This is the first time that vIL-10 variants have been associated to changes in the immune response in HCMV infected individuals.
This study is important, demonstrating the impact of vIL-10 on the HCMV-specific immune response. However, there are still some points in the manuscript that should be clarified.
In the abstract the authors write “Ninety-three percent of samples contained more than one variant of HCMV, as defined by at least one nonsynonymous variation.” It is not clear to me how the amplicons were sequenced. Was the UL111a gene sequenced as one strand or was an assembly step included to cover the whole gene? The following question is, if it can be concluded whether the identified variations from one sample were found on distinct genomes or on one?
In addition, it is mentioned that the variation including a stop codon at position 132 was found in 78% of samples, but how frequent was this change within one sample? Can the authors discuss whether a protein with this premature stop codon could still be stably expressed?
Also regarding the other variants presented in Figures 2, 3 and 4 it is important to know how frequently the variants were detected in the samples.
In Figures 2, 3 and 4 groups are selected that show statistical difference. For completeness, groups with variation in different positions, but with similar number of samples should be included. It is a problem for the statistical evaluation that the tested groups are so small. Especially in Figure 2A this is a problem, since e.g. HLA-A*02 (one of the most frequent HLA-I) positive individuals can have very strong T-cells responses. If, by chance, the three individuals in the D/N+N group have a pp65 responsive haplotype, the high T-cell reactivity might rather be connected to the HLA type and not to the vIL10 variant. With this low number of samples it is not possible to exclude this.
Finally, how were the primers (lines 268-268) selected? What is the chance that a UL111a sequence could be missed due to primer mismatch?
The authors did not discuss splicing events. Could any of the nucleotide changes affect splicing of UL111a?
Table 1 is difficult to understand.
It should be -80 in line 292
Round 2
Reviewer 2 Report
The authors have satisfactorily addressed the points raised in the review. I recommend publication of the manuscript "Sequencing of the viral UL111a gene directly from clinical specimens reveals variants of HCMV encoded IL-10 that are associated with altered immune responses to HCMV".